# Molecular Machinery of Lipid Droplet Degradation and Turnover in Plants

**DOI:** 10.3390/ijms242216039

**Published:** 2023-11-07

**Authors:** Zhaoxia Qin, Tianyu Wang, Yanxiu Zhao, Changle Ma, Qun Shao

**Affiliations:** Shandong Provincial Key Laboratory of Plant Stress, College of Life Sciences, Shandong Normal University, Jinan 250358, China

**Keywords:** lipid droplet, lipolysis, triacylglycerol hydrolysis, lipases, lipophagy

## Abstract

Lipid droplets (LDs) are important organelles conserved across eukaryotes with a fascinating biogenesis and consumption cycle. Recent intensive research has focused on uncovering the cellular biology of LDs, with emphasis on their degradation. Briefly, two major pathways for LD degradation have been recognized: (1) lipolysis, in which lipid degradation is catalyzed by lipases on the LD surface, and (2) lipophagy, in which LDs are degraded by autophagy. Both of these pathways require the collective actions of several lipolytic and proteolytic enzymes, some of which have been purified and analyzed for their in vitro activities. Furthermore, several genes encoding these proteins have been cloned and characterized. In seed plants, seed germination is initiated by the hydrolysis of stored lipids in LDs to provide energy and carbon equivalents for the germinating seedling. However, little is known about the mechanism regulating the LD mobilization. In this review, we focus on recent progress toward understanding how lipids are degraded and the specific pathways that coordinate LD mobilization in plants, aiming to provide an accurate and detailed outline of the process. This will set the stage for future studies of LD dynamics and help to utilize LDs to their full potential.

## 1. Introduction

In plant seeds, stored neutral lipids, such as triacylglycerols (TAGs) and sterol esters, can constitute up to approximately 60% of the dry seed weight [1]. These lipids are stored in cytoplasmic lipid droplets (LDs), which are composed of a hydrophobic matrix of TAGs surrounded by a stable amphipathic membrane [2]. The LD membrane is composed of a phospholipid monolayer and structural proteins, including oleosins, caleosins and steroleosins, which preserve the integrity and stability of LDs [3]. It is worth noting that caleosins were initially described as LD-associated proteins in plant seeds [4]. However, subsequent studies have revealed that caleosins are not only structural proteins but also active peroxygenases and are ubiquitous in plants, fungi and green algae. Caleosin/peroxygenases are a family of multifunctional proteins that also occur in numerous other types of membranes, including the endoplasmic reticulum (ER) and the plasmalemma, and are detected in most plant tissues [5]. Generally, a LD consists of 94–98% (*w*/*w*) TAGs, 0.6-2.0% (*w*/*w*) phospholipids, and 0.6–3.0% (*w*/*w*) protein [6,7]. Seed LDs typically range in diameter from 0.5 to 2 µm and comprise up to ~60% of the volume of a mature embryonic cell. Thus, LDs, along with storage vacuoles, are the most abundant organelles [8]. When needed, stored oil in LDs is degraded and mobilized to provide energy and carbon equivalents for the post-germinative growth within plants [9]. Despite the importance of LD degradation, this process has been less studied in plants compared to animals and yeast and remains only partially understood [10,11].

The primary pathway for mobilizing stored lipids, known as lipolysis, is induced by the activity of endogenous, seedling-specific TAG lipases (TGL), such as the patatin-like lipase SUGAR-DEPENDENT1 (SDP1) in *Arabidopsis* seeds, which rapidly hydrolyzes TAGs to glycerol and free fatty acids (FAs) at the onset of seed germination [12]. The released FA from the LD is subsequently converted to acyl-CoA and degraded via β-oxidation to acetyl-CoA in peroxisomes (glyoxysomes), single membrane-bound organelles that exist in close proximity to LDs and contain three of the five enzymes involved in the glyoxylate cycle (citrate synthase, isocitrate lyase and malate synthase), while the other two enzymes (aconitase and malate dehydrogenase) are present in the cytoplasm [13]. How the released FAs reach or are transported to peroxisomes for further breakdown is not well understood, though some results indicate close physical interactions of LDs with these organelles via peroxules [11]. Subsequently, acetyl-CoA enter the glyoxylate cycle, through which succinate is synthesized. Then, succinate is transported from the peroxisome into the mitochondrion and converted to malate as part of the Krebs cycle. Synthesized malate is transported into the cytoplasm and converted to oxaloacetate. These four-carbon compounds are further metabolized via gluconeogenesis to soluble sugars that are used to fuel seedling growth [1]. Additionally, glycerol is eventually derived from TAG and also feeds into glycolysis or gluconeogenesis [14,15]. How other LD-derived plant neutral lipids, such as sterol esters, are broken down and by which enzymes has been less explored [16].

Before the stored neutral lipids can be mobilized, the LD surface proteins and phospholipid monolayer need to be degraded. During seed germination in *Arabidopsis*, LD surface proteins have been found to be ubiquitinated [8,17,18,19] and removed from the LD membrane to become accessible for degradation by the proteasome [20]. However, little is known about the mechanism of how the phospholipid monolayer is broken down nor the resultant metabolism of the degradation products, although phospholipase activities have been detected in the LD fraction of several plant species [21,22,23].

In plants, another mechanism of LD degradation is lipophagy, the LD-specific form of autophagy through which LDs are selectively delivered to the vacuole for turnover [24]. LDs may be degraded by microautophagy, a kind of autophagy during which LDs are directly engulfed by vacuoles [25]. Although some electron-microscopy-based observations have been made of LDs within vacuoles in different *Arabidopsis* tissues, the role of lipophagy in lipid metabolism and storage in plants has not been well explored [26,27,28]. It is worth noting that pexophagy, autophagic degradation of the peroxisomes, key organelles in lipid mobilization, can also affect LD and lipid mobilization during seed germination in plants. The impact of pexophagy on LD degradation has been confirmed through a series of experiments using embryonic axes isolated from lupin-germinating seeds and cultured in vitro [29]. So far, if and to what extent LDs are taken up into vacuoles and the contribution of lipophagy and pexophagy to LD degradation in plants has remained unclear.

Our focus here is to review recent progress toward understanding how lipids are degraded. We will highlight the specific pathways and proteins that coordinate LD mobilization according to our proposed model for LD degradation in plants (Figure 1). 

## 2. Degradation of the LD Structural Proteins and the Phospholipid Monolayer

Details about the early events of lipid mobilization during germination are still scarce. Several studies have provided some clues about how lipid-mobilizing enzymes, such as TGL and LOX, obtain access to their substrates through the stable LD membrane. Just prior to TAG degradation, the disruption of the integrity of the protein coat and/or the phospholipid monolayer may be required to facilitate or promote these enzymes’ access to the TAGs [30]. Two processes, partial degradation of structural proteins and a transient detected patatin-like phospholipase activity for degrading the phospholipid monolayer were simultaneously observed at the LD membranes during germination [21,31,32].

### 2.1. Proteolytic Degradation of the LD Structural Proteins in Plants

The LD surface proteins are highly dynamic in LD mobilization during seed germination and seedling growth [33]. Some indirect evidence suggests that structural proteins, such as oleosins, protect TAG from lipolytic enzymes. The rate of TAG hydrolysis by exogenous lipase in vitro LDs was slower than that in artificial oil emulsions [34]. Matsui et al. [35] demonstrated that the cucumber LD-specific LOX could catalyze oxygenation of TAGs in vitro only after the structural proteins were partially degraded. In rapeseed seedlings, the partial degradation of oleosin isoforms from 19 kDa to 16 kDa fragments was observed shortly before the complete degradation of LDs [36]. Overexpression of oleosins in yeast or plants prevents lipid turnover [37,38]. In addition, rapid disappearance of oleosins has been observed during postgerminative seedling growth in various plant seeds [39,40,41,42,43]. These results support the general assumption that lipolysis of LDs first requires disruption of the integrity of the oleosin coat. 

Later studies further confirmed that oleosin hydrolysis primarily occurs as the first step of LD mobilization during seed germination and seedling growth via several degradation systems. The reported enzymes involved in this process include the ubiquitin-proteasome-dependent system, the thioredoxin-regulated cysteine protease and the aspartic protease, among others [17,18,19,44,45,46]. 

Degradation of most intracellular proteins occurs via ubiquitination of the respective proteins and subsequent digestion by the proteasome [47]. Ubiquitinated oleosin and caleosin were first detected in sesame after seed imbibition by mass spectrometric analyses. These proteins could be ubiquitinated at the C-terminal lysines [19]. Recently, using forward-genetic screening, MYB30-interacting E3 ligase 1 (MIEL1) was confirmed to ubiquitinate oleosin for degradation in *Arabidopsis thaliana* [48]. MIEL1 is a ubiquitin-protein ligase known to ubiquitinate two MYB transcription factors, MYB30 and MYB96, for degradation during hormone and pathogen responses [49,50]. In the experiments of Traver and Bartel, *miel1* mutants displayed decreased oleosin ubiquitination and slowed TAG degradation, while *MIEL1*-overexpressing seedlings showed low oleosin levels, yielding very large LDs. Surprisingly, MIEL1 localizes to peroxisomes, and *miel1* mutants delay LD mobilization without increasing oleosin transcript levels, suggesting MIEL1 regulates oleosin post-transcriptionally. Transgenic experiments and protein immunoblot suggest that MIEL1 directly ubiquitinates seed oleosins in the vicinity of a peroxisome for subsequent proteasomal degradation during seedling lipid mobilization [48]. 

Proteasomal degradation requires that substrate proteins are removed from the LD membrane [16]. Two research teams revealed the mechanism underlying these ubiquitin-mediated pathways, in which PLANT UBIQUITIN REGULATORY X (UBX)-DOMAIN CONTAINING FAMILY PROTEIN10 (PUX10) and AAA ATPase CELL DIVISION CYCLE 48 (CDC48A) regulate oleosin extraction from the *Arabidopsis* LD monolayer. PUX10 localizes LDs via its hydrophobic domain and binds to the ubiquitinated oleosins. As an adaptor, PUX10 recruits CDC48A to ubiquitinated oleosins and interacts with ubiquitin and CDC48A via its UBA and UBX domains, respectively. Then, oleosins are dislocated from the LDs via the segregase activity of CDC48A [8,18]. Recently, oleosin degradation in *Populus trichocarpa* meristem LDs was also studied. *OLE6*, the most important LD *OLEOSIN* gene, is almost completely downregulated during bud dormancy development, and the oleosins are no longer present. Furthermore, *Populus* homologs of the *Arabidopsis* PUX10 and CDC48A and multiple subunits of the 26S proteasome were identified in the LD fraction. Upregulated expression of PUX10 and the CDC48A isoform during bud dormancy were also observed. The results suggest that these components are involved in removing oleosins from the LDs [51].

During the process of oleosin (OLE) degradation in *Arabidopsis*, OLE1-OLE5, five oleosin isoforms in seed LDs, are hydrolyzed sequentially by proteases, concomitant with several post-translational modifications of the oleosins. OLE5 is degraded first, followed by OLE2 and OLE4, and then OLE1 and OLE3. OLE5 and OLE2 are phosphorylated during LD degradation, while OLE1, OLE3 and OLE4 are ubiquitinated at the onset of lipid degradation. The ubiquitination topology of the various oleosins was complex and different, indicating distinct, specific degradation pathways [17].

Evidence has also been provided for proteolytic digestion of the oleosins [32]. A LD-associated thiol-protease activity in sunflower (*Helianthus annuus* L.) seedling cotyledons, activated by Trx h (thioredoxin h), has been demonstrated to be responsible for oleosin degradation during seed germination. Among the oleosin isoforms (16, 17.5 and 20 kDa) in sunflower seedlings, 17.5 kDa oleosin exhibits maximum susceptibility to protease [52]. As the concentration of protease increased, all oleosin isoforms in LDs were removed [44]. Interestingly, a faster rate of LD mobilization was observed in the presence of polyamine (PA) inhibitors in sunflower seedling cotyledons. Depletion of PA led to reduced retention of some larger oleosins under normal conditions but increased retention of all oleosins under salt stress, indicating that the complex regulation of intracellular PA homeostasis may have an important regulatory role in oleosin degradation and LD mobilization [52]. In soybean, two thiol proteases of the papain family, Bd 30K and P34, were induced during seed maturation and are responsible for the degradation of 24 and 18 kDa oleosins during seed germination [45]. P34-like proteins were also found in the jicama seed [53]. A two-chain aspartic protease identified from the crude extract of peanut LDs showed high affinity for LDs and hydrolyzed both LD intrinsic proteins and extrinsic proteins [46]. These findings indicate that endogenous protease-induced hydrolysis of oleosins may be prevalent in different plants. Moreover, larger molecular weight oleosins were hydrolyzed more quickly than smaller ones [53]. 

### 2.2. The Degradation of the Phospholipid Monolayer

In addition to LD-associated proteins, another component of LD membrane is phospholipids. The phospholipid monolayer of LDs predominantly contains phosphatidylcholine (PC), which accounts for approximately 50% of the total phospholipids. Other components, such as phosphatidylserine, phosphatidylethanolamine and phosphatidylinositol, only account for a small portion [54,55]. 

One hypothesis about how lipases access their substrate TAGs is derived from the observation that phospholipase activity at cucumber cotyledon LDs leads to the formation of holes on the monolayer LD membrane during early stages of germination [31,56]. Phospholipase A2 (PLA2), the patatin-like hydrolase, was first discovered in animals as an enzyme responsible for the hydrolysis of the phospholipid sn-2 fatty acyl bond, releasing free FAs and lysophospholipids [57]. May et al. [58] characterized a clone encoding PLA2 from a cucumber cDNA library prepared from germinating seeds, which coincides with the time of TAG mobilization. It is worth noting that both PLA2 and CsLOX were first detectable after 36 h of germination and continued to increase. CsTGL and CsLOX activity at the LDs increased to a peak level at 96 h of germination, while PLA2 activity decreased considerably at this time point [21]. Immunodetection showed that PLA2 was exclusively confined to LDs [21,58]. Activity analysis using various forms of substrates demonstrated that PLA2 acts on monoacylglycerols and phospholipids and is expressed exclusively during lipid catabolism. PLA2 accumulates after imbibition and onwards, in parallel with declines in oleosin and caleosin levels [21]. Interestingly, the maximal PLA2 expression was simultaneously observed along with holes of 80 nm in width and 2.45 nm in depth in the LD phospholipid monolayer using atomic force microscopy analysis. These holes would be sufficiently large for 100 kDa enzymes, such as LOX and lipases, to access the TAG matrix [31]. Moreover, in vitro evidence showed that the mobilization of storage lipids by LOX was promoted by specific LD phospholipase activity against the phospholipid constituents of the LD membrane [31]. 

In sunflower seedlings, during the early stages of germination, PC is depleted continuously with concomitant accumulation of its degradation product, lysophosphatidylcholine (lyso-PC). Moreover, in vivo PLA2 activity was also detected during this timepoint on the LD surface and was found to be further enhanced in 4- to 5-day old seedlings, coinciding with the phase of maximal lipolysis. This indicates a possible role of PLA2 in the degradation of PC. Confocal laser scanning microscope imaging showed that PLA2 could migrate from the cytosol to the LD surface to facilitate lipolysis [22]. 

Oleosins are phosphorylated during germination [59,60,61]. Interestingly, Parthibane et al. found that OLE3 from immature peanut seeds exhibits both monoacylglycerol acyltransferase and PLA2 activities when overexpressed in yeast and bacterial systems. These bifunctional activities of OLE3 are regulated by phosphorylation, predominantly on serine-18. The GXSXG lipase motif in the N terminus of OLE3 is important for its PLA2 activity. The PLA2 activity of OLE3 leads to decreased phospholipid levels through hydrolysis of PC. Subsequently, the hydrolysis of PC to lyso-PC may disturb the LD membrane structure, inducing the release of TAG from LDs [59,60].

In *Arabidopsis*, phospholipases that catalyze membrane lipid hydrolysis can be divided into acyl-hydrolyzing phospholipase A (PLA) and head group-hydrolyzing phospholipase C (PLC) and phospholipase D (PLD). These enzymes play vital roles in signal transduction during plant growth, development and stress responses [62]. Among these enzymes, four secretory low-molecular-weight PLA2s (sPLA2) (AtPLA2α-δ) have been identified [63]. However, it is currently unclear which phospholipases are involved in degradation of the LD phospholipid monolayer. An inhibitor of PLD activity, 1-butanol, was found to inhibit *Arabidopsis* seed germination accompanied by a retardation in TAG mobilization. Moreover, 1-butanol also induced large-scale RNA degradation in seeds and seedlings, which may lead to abnormal expression of genes necessary for the mobilization of LDs [64], suggesting a possible role of PLD in degradation of the LD membrane. 

Thus far, evidence of phospholipid monolayer degradation by phospholipases has only been found in cucumber, sunflower and *Arabidopsis* [21,22,64]. These studies have suggested that LD degradation is progressive. The loss of LD membrane integrity, consisting of degradation of the structural proteins and phospholipids, initiates LD mobilization at the early stage of seed germination. During this process, phospholipases are recruited to the LD surface and are responsible for the partial degradation of the phospholipid monolayer, thereby making LDs susceptible to lipolytic enzymes, such as TGL and LOX. However, the mechanism behind the migration of PLA2 to the LD surface remains to be explored [65]. 

## 3. Storage Triacylglycerols (TAGs) Mobilization in LDs

### 3.1. SDP1 and SDP1-L Mediated TAG Mobilization

Lipolysis during germination is initiated by TAG hydrolysis catalyzed by endogenous TGLs, which leads to the continuous release of glycerol and energy-rich free FAs, and plays a pivotal role in the plant life cycle [14]. The small size of LDs provides a large surface area per unit of TAG, facilitating lipase binding and lipolysis [66]. In many distant phylogenic plant species, lipase activity is absent in mature seeds and present in seedlings, concomitant with the loss of storage TAGs. SDP1, the major lipase associated with LDs for seed TAG mobilization, was first identified in *Arabidopsis* [12]. SDP1 homologues are distributed widely in plants such as the monocot *Oryza sativa*, the legume *Medicago truncatula*, and the moss *Physcomitrella patens*. SDP1 is responsible for the first hydrolytic attack on the TAG molecule in germinating seeds. The *sdp1* mutant is impaired in lipolysis and exhibits a post-germinative growth arrest phenotype, which can be rescued by external sugar supply [12]. Recently, studies on TAG mobilization in vegetative tissue have also increased dramatically. Compared with wild type, the oil content in the roots and leaves of the sdp1 mutant are increased by 50 times and increased by a slightly lower level, respectively, indicating that lipid breakdown by SDP1 is a limiting factor for oil accumulation in vegetative tissues [67]. 

SDP1 contains a predicted patatin-like domain and shares 74% identity with its homolog SDP1-Like (SDP1-L) in *Arabidopsis*. Among lipases, only SDP1 and SDP1-L, members of an unorthodox class of patatin-like lipases, have been shown to be responsible for hydrolyzing LD TAGs [12,68]. These two proteins are expressed predominantly in seeds, with higher mRNA levels in developing and dry seeds and lower levels following germination. They are also ubiquitously expressed in other vegetative organs. The mRNA level of SDP1L is much lower than that of SDP1 in various tissues [12,69,70]. Together, SDP1 and SDP1L account for more than 95% of TAG hydrolysis in germinating seeds. The germination of mutant *sdp1 sdp1-L* seeds is slowed, and post-germinative seedling growth is severely retarded. Up to 20% of *sdp1 sdp1-L* mutants could finish seedling establishment, and the mutants were not seedling lethal under normal growth conditions [69]. RNA interference suppression of the *SDP1/SDP1-L* gene family resulted in up to an 8% increase of TAGs in oilseed rapeseeds, indicating that seed oil breakdown is mainly attributed to SDP1/SDP1-L [70]. These data also suggest that either other lipases are involved in residual lipid breakdown or alternative energy stores in the form of proteins or soluble sugars may exist, which support post-germinative seedling growth. The oil body lipase 1 (AtOBL1), a homolog of acid lipase RcOBL1 from *Ricinus communis* [71], is localized to LDs and is able to hydrolyze TAGs in *Arabidopsis* seeds. The *obl1* mutant has impaired pollen tube growth in vivo. However, rates of seed germination and TAG breakdown were similar in the mutant to that in wild type plants [72]. Kretzschmar et al. identified a novel LD-associated lipase by a quantitative proteomic analysis during early developmental stages in *Arabidopsis*, but its function has not yet been revealed [73]. 

*Arabidopsis* contains a protein, comparative gene identification-58 (CGI-58), belonging to the α/β-hydrolase superfamily [74]. In mammals, CGI-58 is a key lipase activity modulator localized on LDs, which regulates TAG content and LD abundance. It may help TAG lipase (ATGL) access its substrate TAG for lipid hydrolysis [75]. In *Arabidopsis,* however, CGI-58 is present on peroxisomes in seeds and is a soluble acyltransferase with lipase and phospholipase functions [74]. Disruption of *CGI-58* in *Arabidopsis* resulted in accumulation of TAGs and LDs in mature leaves and yielded a Chanarin–Dorfman syndrome-like phenotype [76,77]. TAG levels in mature *cgi-58* leaves were more than 10-fold higher than that in wild-type plants, while the seed TAG content was unchanged and seeds germinated and grew normally. Mutant *atgl* also showed that ATGLs do not contribute significantly to TAG hydrolysis in *Arabidopsis* seeds [77]. Moreover, *Arabidopsis* CGI-58 recombinant protein expressed in *Escherichia coli* displays TAG lipase activity [74]. Thus, the results suggest that plant CGI-58 could directly participate in the regulation of lipid turnover without requiring an additional TAG lipase on the LD surface in plant nonoil-storing tissues.

In *Arabidopsis*, CGI-58 was discovered to function primarily by interacting with and presumably influencing the activity of PEROXISOMAL ABC-TRANSPORTER1 (PXA1) in leaves. Disruption of CGI-58 (*cgi-58*) resulted in similar increases in TAG and LDs in leaves as that observed in the leaves of *pxa1* mutants [77,78]. PXA1 is a membrane-bound protein that imports a variety of substrates, such as CoA esters of FAs, into peroxisomes for subsequent β-oxidation. In germinating *Arabidopsis* seeds, FAs released by SDP1 are activated with CoA in the cytosol and enter peroxisomes through PXA1, which cleaves the CoA moiety upon transport [79,80]. In the peroxisomes, FAs are reactivated into acyl-CoA by long-chain acyl-CoA synthetases (LACS), such as LACS6 and LACS7, and catabolized to acetyl-CoA through β-oxidation [81]. Thus, PXA1 plays an essential role in TAG turnover in germinated seeds and leaves [77,78]. Notably, CGI-58 is not required for TAG breakdown during seedling establishment [78]. These observations indicate that the interactions between LD and peroxisomes, as well as the way CGI-58 regulates PXA1, may vary between different plant cell types. In *Arabidopsis*, CGI-58 and PXA1 coregulate lipid homeostasis and signaling. 

### 3.2. Delivery of SDP1 from Peroxisomes to LDs

The physical close contact between LDs and peroxisomes plays an important role in the hydrolysis of TAGs [82]. In *Arabidopsis*, SDP1 is originally localized on peroxisomes and then moves to the LD surface [83]. The core retromer vacuolar protein sorting 29 (VPS29) is a protein complex for subcellular cargo trafficking in plants. VPS29 interacts with SDP1 on the peroxisomal surface and mediates the movement of SDP1 from peroxisomes to LDs by regulating peroxisomal tubular extensions for engulfing LDs, although the underlying mechanism is far from clear [67,83]. The endosomal sorting complex required for transport (ESCRT) machinery and PEROXIN 11 (PEX11) are also involved in this process [33]. ESCRT machineries are evolutionarily conserved in eukaryotes and participate in many cellular processes [84]. In *Arabidopsis*, PEX11 proteins play roles in promoting peroxisomal proliferation and tubule formation [85]. The amphipathic helix (AH) motif of PEX11 has membrane remodeling capacity [86,87]. FYVE DOMAIN PROTEIN REQUIRED FOR ENDOSOMAL SORTING 1 (FREE1/FYVE1), a plant-unique ESCRT component, was characterized in *Arabidopsis* [88]. FREE1 interacted directly with the AH motif of PEX11e and SDP1 to promote peroxisomal tubulation and trafficking of SDP1 to the LD surface for TAG mobilization. PEX11e and SDP1 interaction depends on the presence of FREE1, indicating that FREE1 may serve as a scaffold for connecting PEX11e and SDP1 [33]. 

In *Arabidopsis,* small GTPase Rab18 (RABC2a) is localized not only on the LD surface, but is also associated with peroxisomes. Rab8 (RABE1d) is also associated with peroxisomes [89,90]. Moreover, RabC2a interacts with myosin XI (MYA2) for rapid trafficking of peroxisomes [89]. It is speculated that Rab8 and Rab18 might function in LD-peroxisome contact for delivery of lipases from peroxisomal extensions to LDs in *Arabidopsis* [91]. Other than Rab proteins, ADP-RIBOSYLATION FACTOR1 (ARF1) and SECRETION-ASSOCIATED RAS1 (SAR1) may play a role in LD degradation. ARF1 and SAR1 were reported to interact with COPI and COPII vesicles, respectively, to mediate the delivery of ATGL to LD surfaces in mammalian cells [92]. However, their function in LD degradation in plants is still unclear. In *Triadica sebifera*, SAR1, ARF1 and motor protein myosin were detected in isolated LDs from mesocarp and seed kernels [93]. In *Arabidopsis*, the ARF/SAR1 family is involved in various trafficking pathways, such as ER-Golgi traffic, endocytosis and/or recycling and vacuolar trafficking [94]. The possible roles of these proteins in plant LD degradation require further exploration.

### 3.3. Lipid Mobilization by LOX 

In addition to the classical storage lipid degradation pathway by SDP1 and SDP1L, an alternative storage lipid mobilization pathway that is dependent on LD-specific LOX activity has been proposed in several plant species, such as pea, watermelon, cucumber, *Arabidopsis*, wheat, rice, sunflower and *Brassica napus* [95]. LOXs, crucial and ubiquitous enzymes in biological systems, are non-heme, non-sulfur oxidoreductases with iron or manganese as a cofactor. They catalyze the region and stereoselective oxygenation of polyenic FAs, resulting in the corresponding hydroperoxyl derivatives, which are preferentially removed by specific lipases [30]. According to their positional specificity in linoleic acid (LA) oxygenation, plant LOXs may be classified into 9- and 13-LOXs [95]. The activity of 13-LOX leads first to the formation of ester lipid hydroperoxides. For example, it catalyzes hydroperoxidation of either linoleic or linolenic acid residues to (9Z,11E,13S)-13-hydroperoxy octadeca-9,11-dienoic acid (13-HPOD) [96,97,98]. 13-HPOD is subsequently released from LDs and reduced to (9Z,11E,13S)-13-hydroxy octadeca-9,11-dienoic acid (13-HOD), presumably by the peroxygenase activity of caleosin [99,100]. The hydroperoxyl FAs, such as 13-HOD, are then preferentially cleaved off by a TGL and degraded via glyoxysomal β-oxidation [21].

It has been shown that the LOX proteins are expressed at very early stages of germination in cucumber seedlings. Localization analysis by immunogold labeling and transmission electron microscopy demonstrated LOX was directly localized on the LD membrane and also in trace amounts in the cytosol surrounding the LD [21]. In vitro experiments with LD preparations from seedlings showed that large amounts of esterified 13-HPOD are produced by recombinant LD-associated LOX in the membrane fraction. These data suggested that LOX is responsible for oxygenating TAGs, and the lipid mobilization for β-oxidation is initiated by LOX but not by a lipase [21,97]. 

LDs consist primarily of triacylglycerides, in which linoleoyl moieties may constitute more than 60% of the acyl groups in plants [101]. In the LOX pathway, the degradation of storage lipids is initiated by the hydroperoxidation of either linoleic or linolenic acid residues of the TAGs catalyzed by LOX. For example, LA, a major storage FA in many oilseeds [102], may be degraded either directly via β-oxidation or by a LOX-dependent pathway. The two pathways may occur in parallel in plants [103].

## 4. Diacylglycerol (DAG) and Monoacylglycerol (MAG) Mobilization 

During lipolysis, the three fatty acyl chains in a TAG molecule are sequentially hydrolyzed into diacylglycerol (DAG), monoacylglycerol (MAG) and glycerol, releasing a FA molecule at each step. Both SDP1 and SDP1L lipases exhibit the highest hydrolytic activity for TAGs and no detectable activity for DAG or MAG, suggesting that other enzymes catalyze DAG or MAG hydrolysis [12,69]. Diacylglycerol lipases (DAGLs) are widely distributed in bacteria, fungi, plants and animals, albeit with different substrate selectivity. Microbial DAGL can utilize DAG and MAG to produce free FAs, MAG or glycerol, while animal DAGL specifically hydrolyzes DAG and shows very little monoacylglycerol lipase (MAGL) activity [104]. Only 23 putative plant DAGLs were found during a search in the NCBI and Uniprot databases. However, the functions of these enzymes have not yet been identified [105]. A chloroplast non-regioselective lipase, referred to as ADIPOSE TRIGLYCERIDE LIPASE-LIKE (ATGLL), which preferentially removes 16:0 from unsaturated DAGs, was identified to play an important role in maintaining lipid homeostasis in *A. thaliana* [106]. Heterologous expression and enzyme assays indicated that 11 *Arabidopsis* putative monoacylglycerol lipases (MAGLs) indeed possess MAG lipase activity. Among these enzymes, AtMAGL8 has a significant substrate preference for MAG containing characteristic eicosenoic acid (C20:1) and is associated with LDs in germinating seeds and leaves, and is thus a favorable candidate for the MAG lipase involved in seed oil lipolysis [56,107]. However, to date, none of these or any annotated DAG or MAG lipases have been found to work in concert with SDP1 or SDP1-L in LD oil degradation.

## 5. Roles of Autophagy in LD Breakdown in Plants 

### 5.1. Involvement of Autophagy in LD Metabolism in Eukaryotic Cells

Autophagy is a universal mechanism in eukaryotic cells that facilitates the identification and degradation of unwanted or dysfunctional cytoplasmic components. Autophagy is functional at basal levels in all cell types of organisms to maintain cellular homeostasis under favorable growth conditions, whereas the massive inducible autophagy functions primarily as an adaptive response to many developmental and environmental stimuli, such as starvation, senescence, pathogens and other stresses [108,109,110,111]. Based on morphological features, two major types of autophagy-related pathways, microautophagy and macroautophagy, are involved in metabolism and nutrient recycling in eukaryotes [27]. During microautophagy, the tonoplast invaginates to trap cytoplasmic substances, creating autophagic bodies within the vacuole. Macroautophagy is an evolutionarily conserved mechanism in eukaryotic cells in which cytoplasmic constituents are sequestered into double-membrane autophagosomes. The autophagosomes then fuse with the vacuole to release the internal vesicle into the lumen for degradation by lytic enzymes. Macroautophagy is mediated by a group of AUTOPHAGY-RELATED (ATG) proteins that are found in most eukaryotes, including higher plants and animals [112]. 

In addition to lipolysis, autophagy has also been proposed to mediate LD metabolism [113]. Lipophagy involves the membrane engulfment of cytosolic LDs and trafficking to degradation compartments, such as the vacuole or lysosomes, where LD lipids and proteins are degraded [114,115,116]. Lipophagy has been extensively studied in yeast and animals. In mammals, both macrolipophagy and microlipophagy have been reported to be involved in the degradation of LDs [108,113,117,118]. Unlike in mammals, yeast lipophagy resembles microautophagy, and is thus named microlipophagy [119]. However, little is known about the role of autophagy in lipid metabolism in plants [120]. 

### 5.2. Autophagy-Mediated LD Degradation in Plants

Vacuoles in plant cells have storage and lytic functions. In germinating *Arabidopsis* seeds, LDs were primarily observed in the central vacuoles for degradation [121,122]. Analysis of germinating *Atclo1* mutants showed that wild-type cells each have a large central vacuole with a relatively smooth contour and contained LDs, whereas in mutant cells, the vacuoles occasionally contained LDs and had distorted contours, indicating that LDs are broken down via autophagy, and AtCLO1 plays a role in LD–vacuole interactions [26]. Later disruption of autophagy has been found to affect lipid turnover in maize and *Arabidopsis* seedlings [123,124]. In maize, the *atg12* mutant had more LDs in the cytoplasm than in wild-type cells under normal growth conditions, consistent with inhibited autophagic turnover [125]. 

In flowering plants, LD lipids in the tapetum are necessary for pollen maturation and pollen-tube elongation [126]. Kurusu et al. found autophagosome-like structures and numerous vacuole-enclosed LDs in wild-type rice postmeiotic tapetum cells at the uninucleate stage during pollen development. However, these structures were completely abolished in OsATG7 (autophagy-related)-knockout mutant defectives in autophagy, leading to more LDs in the mutant cytoplasm than in the wild-type cells. This suggests that ATG7-dependent autophagy is induced in rice cells and is involved in the breakdown of LDs. Moreover, during tapetal autophagy, LDs directly fuse with the vacuoles, which is distinct from both macro- and microautophagy [127]. 

Fan et al. [27] demonstrated that inducible autophagy in *Arabidopsis* plays a role in LD degradation. Analysis of transgenic plants coexpressing the autophagic marker ATG8e protein and OLE1-GFP revealed that LDs were associated with ATG8e, and LDs were clearly enclosed by the tonoplast under extended darkness-induced starvation conditions. TAG levels in *atg sdp1* double mutants were also increased compared with *sdp1*. Thus, autophagy plays an important role in plant LD degradation, and the process requires the core components of macroautophagy, although its regulatory mechanisms remain largely obscure. 

Barros et al. revealed an altered lipid profile in *Arabidopsis atg5* and *atg7* mutants. The lack of autophagy leads to an imbalance of lipid metabolism and activates the general chloroplast lipid degradation program while failing to produce cytoplasmic LDs in leaves. The results indicate that autophagy is a versatile regulator for the homeostasis of multiple lipid components and LD metabolism, particularly during extended dark-induced senescence [128]. 

In addition to lipophagy, pexophagy may also play an important role in the degradation of LDs in plants. Under nutrient deficient conditions, the mobilization of storage lipids in LDs is enhanced to provide energy and carbon equivalents for post-germinative growth [9]. However, under sugar starvation conditions, it was observed that there was significantly higher lipid content in isolated lupin embryonic axes cultured in vitro, indicating that lipid breakdown was disrupted. Further experiments confirmed that enhanced pexophagy could partially explain the results. Moreover, ultrastructure observation showed that lipophagy was also restricted under starvation conditions [29].

Although recent studies have clearly revealed the interaction between autophagy and lipid metabolism [27,123,125,129], whether autophagy mediates the biogenesis or degradation of LDs in vegetative tissues remains controversial [128,130,131].

### 5.3. Possible Proteins Involved in Autophagy-Mediated LD Degradation in Plants 

As the core protein of autophagy, ATG8 interacts with numerous adaptors and receptors, usually containing the ATG8-interacting motif (AIM), to help generate and transport autophagic vesicles [132,133,134]. AIM binds an AIM docking site (LDS) on ATG8e [135,136]. Marshall et al. [137,138] found a new class of ATG8-interactors that use ubiquitin-interacting motif (UIM)-like sequences for high-affinity binding to the UIM-docking site (UDS) on ATG8 in *Arabidopsis*. Thus, a single ATG8 could dock UIM- and AIM-containing proteins simultaneously. Four PUX proteins (PUX7, PUX8 PUX9 and PUX13) from *Arabidopsis* displayed robust interactions with ATG8 using UIM motifs and were also found to interact with CDC48. The UIM–UDS contact is critical for autophagic clearance of dysfunctional CDC48/p97 complexes. Deruyffelaere et al. [18] proposed that PUX10, another PUX protein, localizes to LDs and recruits CDC48A to the ubiquitinated oleosins to dislocate oleosins from LDs. Whether ATG8 can interact with PUX10 and CDC48A for the degradation of oleosins and LDs through autophagy remains to be determined. 

Studies have implied that in *Arabidopsis*, FREE1 also regulates autophagosome–vacuole fusion and subsequently autophagic degradation via direct interaction with a unique plant autophagy regulator SH3 DOMAIN-CONTAINING PROTEIN2 (SH3P2) [139]. However, it remains to be explored whether the ESCRT components play a direct role in plant LD degradation [91].

Data suggest that Rab proteins, such as Rab32, Rab7 and Rab10, may participate in LD degradation through autophagy in mammalian cells [140,141,142,143,144]. In plants, Rab GTPases have been widely studied and reported to play a role in various membrane-trafficking pathways [145]. There are eight putative Rab7 homologs in the *A. thaliana* genome [146]. Rab7 homolog (RABG3f) proteins are localized to the tonoplast, and proper activation of RABG3f is important for both vacuole biogenesis and prevacuolar compartments (PVCs)-to-vacuole trafficking [147]. In addition, disruption of RABG3f activity blocked fusion between autophagosomes and the vacuole membrane, thereby disturbing autophagy [148]. Experiments revealed that *Arabidopsis* dimeric complex, formed by MONENSIN SENSITIVITY1 (SAND/MON1) and CALCIUM CAFFEINE ZINC SENSITIVITY1 (CCZ1), serves as the RABG3f guanine nucleotide exchange factor (GEF) and is critical for the fusion of multivesicular endosomes (MVEs) with the vacuole, and thus is also important for plant growth [149,150]. However, the detailed function of RABG3f and its role in LD degradation through autophagy in plants are still unknown.

## 6. Role of the LD Degradation in Pathogen Infection in Plants

More recently, preliminary research has revealed a relationship between LD degradation and pathogen infection in plants. When exposed to pathogen infection, plants can induce massive intricate changes of host primary metabolism, including lipid metabolism [151,152]. As the source of lipids and cholesterol, LDs are unsurprisingly frequently co-opted by a variety of pathogens for their growth and proliferation in mammalian cells [153]. Many viruses, such as hepatitis C virus (HCV), can disrupt LD degradation to facilitate their own replication systems [154]. 

In some plants, LD biosynthesis and TAG accumulation are induced in response to pathogens (reviewed in [155]). However, little is known about the role of LD degradation in plant pathogen infection. Laufman et al. [156] demonstrated that poliovirus recruits LDs to the forming viral organelles to obtain products of LD lipolysis—FAs—and transport them to viral replication compartments (VRCs). When potato is infected with the oomycete *Phytophtora infestans*, the LD degradation in leaf guard cells is induced to maintain stomatal opening, thereby promoting the emergence of sporangiophores from the opened stomata [157]. Wang et al. [158] found that rice black-streaked dwarf virus (RBSDV) infection leads to increased accumulation of C18 polyunsaturated fatty acids (PUFAs) in maize. Further experiments showed that overexpression of the LD-associated protein (ZmLDAP1 or ZmLDAP2), the main structural LD proteins in non-seed tissues [159], induces LD clustering. The core capsid protein P8 of RBSDV associates with the clustered LDs, interacts with the LD-associated proteins (ZmLDAP2) and prevents its degradation via the PUX10-mediated ubiquitin–proteasome system. Thus, RBSDV interferes with the degradation of LD related proteins to regulate the metabolism of C18 PUFAs, thereby promoting virus replication and infection [158]. Further investigation is needed to elucidate the details of interaction between LD degradation and pathogen infection in plants.

## 7. Concluding Remarks and Future Perspectives

In plants, seed germination is the beginning of the life cycle and is characterized by the mobilization of stored lipids in LDs. LD mobilization is a complex process involving many proteins, and the detailed mechanism in different species may vary. Although emerging findings, most of which derive from studies in animal adipocytes, have greatly expanded our current understanding of the process in eukaryotic cells, the basic principles controlling LD turnover are still largely unknown, especially in plants. To date, some key proteins involved in LD degradation in plants have been identified (Table 1). Their mechanisms of action have only just begun to be revealed, and some of the functions of these proteins are unclear. Many interesting questions need to be further investigated. For example, how lipolytic enzymes are targeted to the LD surface during LD degradation are still unclear. Little is known about the complex cross-talk between LDs and other compartments, such as peroxisomes. How LDs are recognized by the autophagic machinery and how lipophagy is regulated, as well as the relative contribution of lipolysis and lipophagy for LD turnover, need to be explored. More candidate proteins and genes involved in LD degradation are waiting for further verification. Using a combination of in vitro experiments based on LDs or artificial emulsion droplets and cell biology approaches, such as genetic screens, proteomics, mechanistic studies of the candidate proteins, as well as technological improvements, will enable addressing these important questions. Lipids stored in LDs play crucial roles in the growth and development of plants. A deeper and more comprehensive understanding of these questions will lay a foundation for further establishing a complete and accurate model of the mechanisms underlying lipid and LD mobilization.

## Figures and Tables

**Figure 1 ijms-24-16039-f001:**
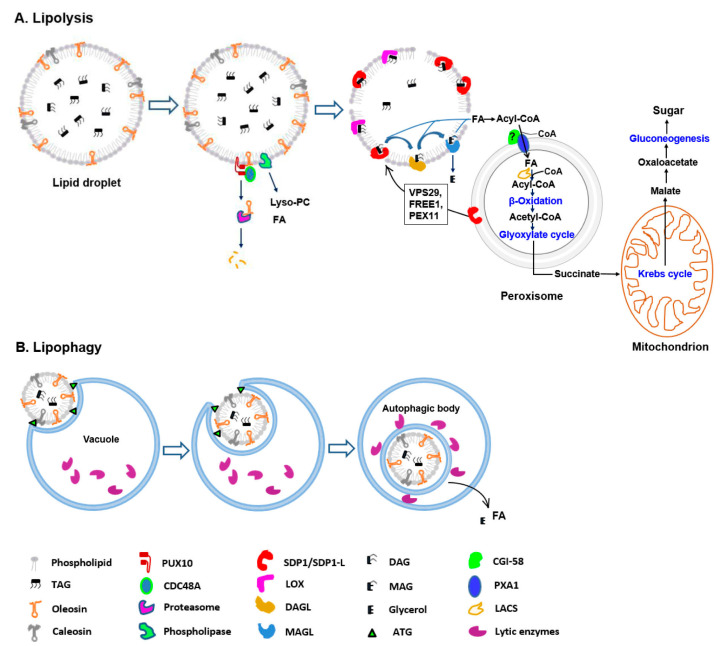
Model of lipid droplet degradation during seed germination in plants. (**A**) Lipolysis. Prior to TAG degradation, the structural proteins and LD phospholipid monolayer are partially degraded mainly via the ubiquitin–proteasome-dependent system and phospholipase activities, respectively. Subsequent TAG mobilization is catalyzed by lipid-mobilizing enzymes, such as SDP1 and lipoxygenase (LOX). The released FAs are transported into peroxisomes for β-oxidation and converted to acetyl-CoA. Next, acetyl-CoA enters the glyoxylate cycle, through which succinate is synthesized. Then, succinate is transported into the mitochondrion and converted to malate through a part of the Krebs cycle. Finally, soluble sugars are synthesized via gluconeogenesis. (**B**) Lipophagy. The process may occur by vacuolar membrane invagination, which increases, and LD is engulfed. Inside the vacuole, a single bilayer membrane autophagic body containing LD is released. Finally, the autophagic body is degraded and lipids in LDs are hydrolyzed into FAs and glycerol.

**Table 1 ijms-24-16039-t001:** Identified proteins involved in lipid droplet degradation in *Arabidopsis thaliana*.

Protein	TAIR Locus	Localization	Putative Function	Reference
PUX10	AT4G10790	LD	Acts as an adaptor to recruit CDC48A to ubiquitinated oleosins	[8,18]
CDC48A	AT3G09840	LD	AAA-type ATPase that facilitates oleosin dislocation from LDs	[8,18]
MIEL1	AT5G18650	Peroxisome	Targets and ubiquitinates seed oleosins for degradation during seedling lipid mobilization	[52]
PLA2	AT2G06925 AT2G19690 AT4G29460 AT4G29470	Cytosol	Hydrolysis of the sn-2 fatty acyl bond of phospholipids, involved in lipid catabolism and phospholipid metabolisms	[62,63]
PLD α1	AT3G15730	Intracellular membranes	Hydrolysis of membrane phospholipids	[160]
OLE1	AT4G25140	ER, LD	Might be involved in lipid metabolism	[161]
OLE3	AT5G51210	LD	Exhibits both monoacylglycerol acyltransferase and PLA2 activities	[59,60]
SDP1	AT5G04040	Peroxisome	Hydrolyze LD TAGs	[12]
SDP1-L	AT3G57140	Unknown	Hydrolyze LD TAGs	[12]
AtOBL1	At3G14360	LD	Hydrolyze LD TAGs	[72]
LOX	AT1G17420	Cytosol	Catalyzes the oxygenation of FAs	[21,101]
ATGLL	AT1G33270	Chloroplast	Catalyzes DAG hydrolysis and plays important roles in maintaining lipid homeostasis in plants	[106]
MAGL 8	AT2G39420	ER and/or Golgi network	Associated with the LD surface and catalyzes MAG hydrolysis during seed germination	[107]
CGI-58	AT4G24160	Peroxisome	Soluble acyltransferase, with lipase and phospholipase functions; participates in the regulation of lipid turnover	[77]
PXA1	AT4G39850	Peroxisome	Imports polyunsaturated FAs into peroxisomes for β-oxidation	[78]
LACS6	AT3G05970	Peroxisome	Activates fatty acids into long-chain acyl-CoA for further β-oxidation in peroxisomes.	[81]
LACS7	AT5G27600	Peroxisome	Activates fatty acids into long-chain acyl-CoA for further β-oxidation in peroxisomes.	[81]
VPS29	AT3G47810	MVBs	Mediates peroxisome tubulation to deliver the lipase SDP1 to the LD surface	[83,162]
FREE1	AT1G20110	Peripheral membrane	Mediates the movement of SDP1 from peroxisomes to LD	[88]
PEX11e	AT3G61070	Peroxisome	Mediates the movement of SDP1 from peroxisomes to LD	[33]
RABE1d	AT5G03520	Peroxisome	Small GTPases which might promote LD turnover	[90]
RABC2a	AT5G03530	LDs, peroxisome	Small GTPases which might promote LD turnover	[89]
ARF1	AT1G23490	Cytosol, endomembrane	Might play a role in the delivery of ATGL to the LD surface for lipolysis	[94]
SAR1	AT1G56330	ER, cytosol,	Might play a role in the delivery of ATGL to the LD surface for lipolysis	[94]
CLO1	AT4G26740	LD	Regulates the normal modification of the vacuole membrane and the interaction of LDs with vacuoles	[26]
RABG3f	AT3G18820	PVCs, tonoplast	Small GTPases that may promote direct interactions between MVBs/lysosomes and LDs	[147]
CCZ1	AT1G16020	PVCs	Vacuolar fusion protein	[149]
SAND/MON1	AT2G28390	PVCs, tonoplast	Essential for the fusion of MVBs with the vacuole	[149]
SH3P2	AT4G34660	autophagosome membrane	Critical non-ATG regulator of plant autophagy	[139]
ATG2	AT3G19190	autophagosome precursors	Participate in phagophore expansion and LD breakdown	[163]
ATG5	AT5G17290	Autophagosomes	Involved in ATG8 lipidation in autophagy and LD breakdown	[128]
ATG7	AT5G45900	Cytosol	E1 enzyme for ATG12 and ATG8-family proteins in autophagy and involved in breakdown of LDs	[163]
ATG8e	AT2G45170	Autophagosomes	Ubiquitin-like protein that decorates the phagophore via conjugation to phosphatidylethanolamine and involved in breakdown of LDs	[27]
ATG12	AT1G54210	Autophagosomes	Involved in ATG8 lipidation in autophagy and LD breakdown	[125]

PLDα1, Phospholipase Dα1; MVB, multivesicular bodies; CLO1, caleosin1.

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
