# Peer review of "Molecular Machinery of Lipid Droplet Degradation and Turnover in Plants"

_ijms, 2023, doi:10.3390/ijms242216039_

Round 1
Reviewer 1 Report
Comments and Suggestions for Authors
The manuscript contains two sets of data related to lipid droplet degradation in plants. One is related to lipolysis and the second one is to autophagy (lipophagy). Unfortunately, I can not recommend the acceptance of the manuscript due to the following reasons.
1. The manuscript contains many ambiguities and serious mistakes. Most of them I have marked in the PDF and indicated sources that should be considered by the authors. Also, the digressions to mammalian data are rather not necessary and not understandable, because the manuscript is related only to plants. Some data are also available from experiments conducted on yeasts, but none of such data are in the manuscript. An example can be Atg15 possessing lipolytic activity and its function is quite good described in yeasts. So, if the mammalian data are referred to, why nothing is about yeasts?
2. Several original and review papers related to lipid droplet degradation in plants were published in the recent few years. Unfortunately, the merit level of these papers is remarkably higher than the manuscript. Below I listed some of them. Unfortunately, none of them is cited in the text. Authors should update the text and ensure that the newest data are included in their review paper.
https://academic.oup.com/jxb/article/73/9/2848/6502462
https://www.mdpi.com/2223-7747/11/9/1243
https://www.frontiersin.org/articles/10.3389/fpls.2020.579019/full
https://www.frontiersin.org/articles/10.3389/fpls.2021.674031/full
https://www.mdpi.com/1422-0067/24/14/11773

Author Response
Response to Reviewer 1 Comments
We thank the reviewer for their comments. According to the comments of the reviewer, we have carefully read the relevant papers on plant LD degradation and corrected the errors in the article and Figure1. We have also carefully reviewed some new articles and incorporated the newest data into our review paper with the relevant citations. All changes are in red.
The manuscript contains two sets of data related to lipid droplet degradation in plants. One is related to lipolysis and the second one is to autophagy (lipophagy). Unfortunately, I can not recommend the acceptance of the manuscript due to the following reasons.
Point 1: The manuscript contains many ambiguities and serious mistakes. Most of them I have marked in the PDF and indicated sources that should be considered by the authors. Also, the digressions to mammalian data are rather not necessary and not understandable, because the manuscript is related only to plants. Some data are also available from experiments conducted on yeasts, but none of such data are in the manuscript. An example can be Atg15 possessing lipolytic activity and its function is quite good described in yeasts. So, if the mammalian data are referred to, why nothing is about yeasts?
Response 1:
For each section pointed out by the reviewer, we have carefully read the relevant papers and have revised, reorganized and supplemented each section as appropriate.
Compared to animals, understanding of LD degradation in plants is more fragmented and the mechanistic details of the process remain poorly defined. Many proteins have been shown to participate in LD degradation in animals. Homologous proteins exist in plants; however, their functions are still unclear. Thus, it's a bit difficult to manage the story of the subject purely from a plant point-of-view. We aimed to have a better discussion on the possible functions of these plant proteins in LD degradation by comparing them with animal data, but as pointed out by the reviewer, this could be improved. According to the comments of the reviewer, we have done our best to reorganize the relevant literature and made some modifications, retained necessary data and removed parts unrelated to plants. Moreover, due to limitations in length and word count, we mainly compared higher animals and plants, rather than yeast, to clarify the story of the review.
Point 2: Several original and review papers related to lipid droplet degradation in plants were published in the recent few years. Unfortunately, the merit level of these papers is remarkably higher than the manuscript. Below I listed some of them. Unfortunately, none of them is cited in the text. Authors should update the text and ensure that the newest data are included in their review paper.
https://academic.oup.com/jxb/article/73/9/2848/6502462
https://www.mdpi.com/2223-7747/11/9/1243
https://www.frontiersin.org/articles/10.3389/fpls.2020.579019/full
https://www.frontiersin.org/articles/10.3389/fpls.2021.674031/full
https://www.mdpi.com/1422-0067/24/14/11773
Response 2:
We thank the reviewer for listing the most recent relevant articles about LD degradation in plants. We have carefully read all of them and other new articles. We have incorporated the newest data into our review paper with the relevant citations.
The following is responses to the comments of the reviewer in the PDF version:
Point 3: This is misleading and not clear because autophagy always engages vacuole. Macroautophagy (autophagosome) and microautophagy (vacuole) should be added. Response 3:
We have modified the sentence as “lipophagy in which LDs are degraded by autophagy”
Point 4: Keywords should not be the same as words already used in a title.
Response 4:
We have modified the keywords as “lipid droplet; lipolysis;”.
Point 5: Not true. Please check Borek et. al 2015 Acta Physiol Plant (2015) 37:119
DOI 10.1007/s11738-015-1871-2
Response 5:
We have carefully reviewed the paper and corrected the data in the text in line 27. The reference1 has been added in the References section.
Point 6: Not only oilseed plants. Storage lipids are important for post-germinative growth also in other plants. In protein and carbohydrate-storing seeds as well.
Response6
According to the comments of the reviewer, “oilseed plants” has been modified to “plants” in line 42.
Point 7: “comparatively to what?”
Response7
We have modified the sentence to “Despite the importance of LD degradation, this process has been less studied in plants compared to animals and yeast and remains only partially understood” in line 43-44
References have been added (Reference 10 and 11).
Point 8: I cannot agree. Please refer to Borek et al. 2015 and references cited there. Acta Physiol Plant (2015) 37:119 DOI 10.1007/s11738-015-1871-2
Response 8:
According to the comments of the reviewer, we have carefully reviewed the paper and modified the relevant content in the text in line 51-60. References have been added (Reference1 and 13).
Point 9: Fatty acids also are carbon compounds.
Response 9:
Yes, I agree. Modifications have been made here, see Response 8.
Point 10: add 'peroxules' here and cite Choi, Y.J.; Zaikova, K.; Yeom, S.-J.; Kim, Y.-S.; Lee, D.W. Biogenesis and Lipase-Mediated Mobilization of Lipid Droplets in Plants. Plants 2022, 11, 1243. https://doi.org/10.3390/plants11091243
Response 10:
We have added 'peroxules' in line56 and cited the paper (Reference11).
Point 11: Between the glyoxylate cycle and gluconeogenesis, a part of the TCA cycle is needed. Borek et al. 2015.
Response 11:
We have supplemented the relevant content of TCA cycle (Krebs cycle) into the text and cited the paper (Reference1).
Point 12. Macroautophagy is a kind or type of autophagy; not a component!
Response 12:
Yes,I agree. We have revised it as “a kind of autophagy” in line74.
Point 13. Also, pexophagy should be taken into consideration here. Please chek Borek, S.; Stefaniak, S.; Nuc, K.; Wojtyla, Ł.; Ratajczak, E.; Sitkiewicz, E.; Malinowska, A.; ´Swiderska, B.; Wleklik, K.; Pietrowska-Borek, M. Sugar Starvation Disrupts Lipid Breakdown by Inducing Autophagy in Embryonic Axes of Lupin (Lupinus spp.) Germinating Seeds. Int. J. Mol. Sci. 2023, 24, 11773. https://doi.org/10.3390/ijms241411773.
Response13:
According to the comments of the reviewer, we have supplemented the relevant content of pexophagy in line77-82 and line 461-468and cited the paper (Reference29).
Point 14: but B-oxidation is not the end of storage lipid conversion into sugars. (figure1)
Response 14:
According to the comments of the reviewer, we have added the following process of oxidation to sugar formation in the figure1.
Point 15: This part contains serious and substantive mistakes!!! Please count layers of phospholipids in the autophagosome and vacuole. In autophagosome must be 4, not 2. The autophagic body is missing - each cargo of autophagosome is initially surrounded by a single phospholipid membrane and creates an autophagic body inside the vacuole. Please check Stefaniak et al. 2020. Int. J. Mol. Sci. 2020, 21, 2205; doi:10.3390/ ijms21062205(figure1)
Response 15:
According to the comments of the reviewer, we have carefully reviewed the paper and concluded that the LD degradation in plants is through the process of microautophagy. We have modified the process in figure1.
Point 16: Not LDs are released into the lumen of the vacuole, but inside the vacuole the autophagic body is created which contains LDs inside. (figure1).
Response 16:
According to the comments of the reviewer, we have carefully reviewed the paper and added the autophagic body which contains LDs inside the vacuole.
Point 17: Not marked in the picture.
Response 17:
Because steroleosin is not the main protein involved in the text and can be omitted. It has been deleted in the figure.
Point 18: Each abbreviation must be explained when is used for the first time(figure1).
Response 18:
According to the comments of the reviewer, the abbreviation “LOX” has been explained.
Point 19: What is the connection between these data and plants?
Response 19:
We think that comparing data from animals will help discuss the possible functions of homologous proteins in plants. According to the comments of the reviewer, we modified the text, retained necessary data and removed parts unrelated to plants.
Point 20: Not related to plants.
Response 20:
According to the comments of the reviewer, we have modified the text, and removed parts unrelated to plants.
Point 21: In addition to lipophagy, pexophagy may also be important in the degradation of LDs in plants. Borek, S.; Stefaniak, S.; Nuc, K.; Wojtyla, Ł.; Ratajczak, E.; Sitkiewicz, E.; Malinowska, A.;´Swiderska, B.; Wleklik, K.; Pietrowska-Borek, M. Sugar Starvation Disrupts Lipid Breakdown by Inducing Autophagy in Embryonic Axes of Lupin (Lupinus spp.) Germinating Seeds. Int. J. Mol. Sci. 2023, 24, 11773. https://doi.org/10.3390/ijms241411773
Response 21:
We have carefully reviewed the paper and supplemented the relevant content of pexophagy in line77-82 and line 461-468 and cited the paper (Reference29).
Point 22: “LC3 (mammalian homologue of ATG8)” (line 437 in PDF): highlight annotation.
Response 22:
“LC3 (mammalian homologue of ATG8)” and LC3- relevant content has been deleted.
Point 23: Content of Line 456-462 in PDF: highlight annotation
Response 23:
We have modified the text, and deleted parts unrelated to plants.
Point 24. Line 479 in PDF: “LD mobilization is a complex process involving many different proteins with the detailed mechanism differing in different species.”: highlight annotation.
Response 24:
The sentence has been modified as “LD mobilization is a complex process involving many proteins, and the detailed mechanism in different species may vary.”
Point 25: Line 482-483 in PDF: “To date, only a few homologs of the key proteins involved in LD degradation in animals have been identified in plants”: highlight annotation.
Response 25:
The sentence has been revised to” To date, some key proteins involved in LD degradation in plants have been identified.”
Point 26: Line 496-497: “A deeper and more comprehensive understanding of LD degradation will lay a foundation for cultivating new crop varieties with high oil yield and improving the industrial application value of LDs” : highlight annotation.
Response 26:
The sentence has been revised to” A deeper and more comprehensive understanding of these questions will lay a foundation for further establishing a complete and accurate model of the mechanisms underlying Lipid and LD mobilization.”
Point 27: An additional section 'Abbreviations' should be added which will contain all abbreviations used in the text.
Response:
We have explained each abbreviation when is used for the first time in the text.

Reviewer 2 Report
Comments and Suggestions for Authors
In this manuscript, Qin et al have very succinctly summarized the mechanism of lipid degradation in plants. While this manuscript is sufficiently summarizing the current knowledge in the field, the missing pieces of how LDs are mobilised are highlighted but not given any possible hypothesis. This manuscript will definitely benefit from a proposed mechanism of what the authors think could be involved in this process. The authors can also draw parallels from other organisms if possible.
Additionally, are there any plan pathogens that co-opt this lipid degradation mechanism during infection. A small section on this could greatly enhance the impact of this manuscript.
Author Response
Response to Reviewer 2 Comments
Thanks for the comments of the reviewer very much. All changes are in red.
Point 1: In this manuscript, Qin et al have very succinctly summarized the mechanism of lipid degradation in plants. While this manuscript is sufficiently summarizing the current knowledge in the field, the missing pieces of how LDs are mobilised are highlighted but not given any possible hypothesis. This manuscript will definitely benefit from a proposed mechanism of what the authors think could be involved in this process. The authors can also draw parallels from other organisms if possible.
Response 1:
Compared to animals, understanding of LD degradation in plants is more fragmented and the mechanistic details of the process remain poorly defined. Many proteins have been proven to participate in the LD degradation in animals. Homologous proteins exist in plants, however, their functions are still unclear. It's a bit difficult to propose a complete hypothesis of the subject. According to the comments of the reviewer, we have also consulted some latest relevant literature and made some revisions, reorganizations, and supplements into the text, such as relevant content in line 51-60, line 77-82, line 150-157, etc. On this basis, we attempted to speculated on the entire process of plant LD degradation and made modifications to Figure 1.
Due to due to limitations in length and word count, we mainly compared higher animals and plants, and supplemented some data from animals to clarify the story of the review.
Point 2: Additionally, are there any plant pathogens that co-opt this lipid degradation mechanism during infection. A small section on this could greatly enhance the impact of this manuscript.
Response 2:
We think this is a very good suggestion. We have consulted some latest relevant literature about the interaction between LD degradation and pathogen infection. The relevant content is supplemented as part 6 in the text. References have also been updated.

Reviewer 3 Report
Comments and Suggestions for Authors
Molecular Machinery of Lipid Droplet Degradation in Plants
In this review article the authors provide a timely analysis of some of the mechanisms that are associated with lipid droplet (LD) degradation. The review is basically sound but has several deficiencies that need to be corrected before it can be accepted after resubmission.
· A major issue is that the title only mentions degradation, which implies that the LDs are simply stores of lipid that are irreversibly broken down to release their contents at some point in plant development, eg after seed or pollen germination. However, the authors quite correctly also discuss LD turnover, which is a more dynamic concept whereby LD contents might participate in a dynamic two-way exchange with other cellular compartments. One example would be the temporary sequestration and release of lipids from or to cell membrane systems including thylakoids or plasmalemma. Such processes are universally seen in other eukaryotes from yeast to mammals.
· Hence, turnover should be included in the title.
· There are some useful comparisons with mammalian systems, which are probably the best characterised and these would be usefully supplemented by data from this recent article:
· Mammalian lipid droplets: structural, pathological, immunological and anti-toxicological roles, Progress in Lipid Research 91,101233, https://www.sciencedirect.com/science/article/pii/S0163782723000231
· In line 129 the authors mention a study by Traver & Bartel but no citation is given – presumably they mean this 2023 paper, which should be discussed in more detail.
· The ubiquitin-protein ligase MIEL1 localizes to peroxisomes to promote seedling oleosin degradation and lipid droplet mobilization
· https://www.sciencedirect.com/science/article/pii/S0163782723000231\
· Caleosins are repeatedly described as ‘structural proteins’. This might be partially correct in some cases but it is definitely not true in many cells and membranes. In fact caleosins are now known to be active peroxygenases that are found in plants and many fungi where they occur on numerous types of membranes as well as on LDs. The authors should rectify this by including a proper factual discussion about caleosins and their putative roles on LDs etc. A recent review can be found at: Plant caleosin/peroxygenases (CLO/PXG), Annals Botanyhttps://academic.oup.com/aob/advance-article-abstract/doi/10.1093/aob/mcad001/6992885?utm_source=advanceaccess&utm_campaign=aob&utm_medium=email
Free-access
OK but minor errors
Author Response
Response to Reviewer 3 Comments
In this review article the authors provide a timely analysis of some of the mechanisms that are associated with lipid droplet (LD) degradation. The review is basically sound but has several deficiencies that need to be corrected before it can be accepted after resubmission.
Point 1: A major issue is that the title only mentions degradation, which implies that the LDs are simply stores of lipid that are irreversibly broken down to release their contents at some point in plant development, eg after seed or pollen germination. However, the authors quite correctly also discuss LD turnover, which is a more dynamic concept whereby LD contents might participate in a dynamic two-way exchange with other cellular compartments. One example would be the temporary sequestration and release of lipids from or to cell membrane systems including thylakoids or plasmalemma. Such processes are universally seen in other eukaryotes from yeast to mammals.
- Hence, turnover should be included in the title.
Response 1:
“turnover” has been has been added to the title and marked in red.
Point 2: There are some useful comparisons with mammalian systems, which are probably the best characterised and these would be usefully supplemented by data from this recent article:
Mammalian lipid droplets: structural, pathological, immunological and anti-toxicological roles, Progress in Lipid Research 91,101233, https://www.sciencedirect.com/science/article/pii/S0163782723000231
Response 2:
According to the comments of the reviewer, we have carefully reviewed the paper. We also read some other papers and modified the relevant content in the part 6 of the text. References have been added.
Point 3: In line 129 the authors mention a study by Traver & Bartel but no citation is given – presumably they mean this 2023 paper, which should be discussed in more detail.
- The ubiquitin-protein ligase MIEL1 localizes to peroxisomes to promote seedling oleosin degradation and lipid droplet mobilization
- https://www.sciencedirect.com/science/article/pii/S0163782723000231\
Response 3:
Thanks for the reviewer very much. We have added the references in the text. We have carefully reviewed the paper and discussed in more detail about the data of this paper in line 130-141 in red.
Point 4: Caleosins are repeatedly described as ‘structural proteins’. This might be partially correct in some cases but it is definitely not true in many cells and membranes. In fact caleosins are now known to be active peroxygenases that are found in plants and many fungi where they occur on numerous types of membranes as well as on LDs. The authors should rectify this by including a proper factual discussion about caleosins and their putative roles on LDs etc. A recent review can be found at: Plant caleosin/peroxygenases (CLO/PXG), Annals Botany https://academic.oup.com/aob /advance-article-abstract/doi/10.1093/aob/mcad001/6992885?utm_source=advance access&utm_campaign=aob&utm_medium=email Free-access
Response 4:
We have rectified the definition of caleosins and added the discussion in line 31-37.
Line 66: “LD proteins, including oleosins, caleosins and steroleosins” has been revised to “LD surface proteins”.
Line 363: “the LD protein caleosin” has been revised to “caleosin”.

Round 2
Reviewer 1 Report
Comments and Suggestions for Authors
Initially, I was critical of the manuscript, but the revised version I have read with pleasure. Now, I can only thank the authors that so considerably improved the text. All of my objections were considered by the authors and point-by-point changes are introduced to the text. It is worth adding that not only my comments were taken under consideration but other changes were also introduced. A new chapter 6 (Role of the LD Degradation in Pathogen Infection in Plants) was added to the manuscript. This new section suits very well to the topic of the manuscript and is a valuable addition and elevates the overall merit evaluation of the manuscript. Currently, I have no objection to the text and can only recommend acceptance of the manuscript.